# Mycelium Polysaccharides from *Termitomyces albuminosus* Attenuate CCl_4_-Induced Chronic Liver Injury Via Inhibiting TGFβ1/Smad3 and NF-κB Signal Pathways

**DOI:** 10.3390/ijms20194872

**Published:** 2019-09-30

**Authors:** Huajie Zhao, Huaping Li, Yanbo Feng, Yiwen Zhang, Fangfang Yuan, Jianjun Zhang, Haixia Ren, Le Jia

**Affiliations:** 1Institute of Agricultural Resources and Environment, Shandong Academy of Agricultural Science, Jinan 250100, China; zhaohuajiebest@163.com; 2College of Life Science, Shandong Agricultural University, Taian 271018, China; 18854890971@163.com (H.L.); fungyimbo@163.com (Y.F.); zhangyiwenwa@163.com (Y.Z.); 18815388639@163.com (F.Y.); zhangjj@sdau.edu.cn (J.Z.)

**Keywords:** anti-oxidation, anti-inflammatory, anti-fibrosis, liver injury, mycelium polysaccharides, *Termitomyces albuminosus*

## Abstract

A major fraction (MPT-W), eluted by deionized water, was extracted from mycelium polysaccharides of *Termitomyces albuminosus* (MPT), and its antioxidant, anti-fibrosis, and anti-inflammatory activities in CCl_4_-induced chronic liver injury mice, as well as preliminary characterizations, were evaluated. The results showed that MPT-W was a polysaccharide of α- and β-configurations containing xylose (Xyl), fucose (Fuc), mannose (Man), galactose (Gal), and glucose (Glc) with a molar ratio of 0.29:8.67:37.89:35.98:16.60 by gas chromatography-mass spectrometry (GC-MS), Fourier transform infrared (FT-IR) spectroscopy. Its molecular weight (Mw), obtained by high-performance gel permeation chromatography (HPGPC), was 1.30 × 10^5^ Da. The antioxidant assays in vitro showed that MPT-W displayed scavenging free-radical abilities. Based on the data of in vivo experiments, MPT-W could inhibit TGFβ1/Smad3 and NF-κB pathways; decrease the level and activity of cytochrome P4502E1 (CYP2E1), malonaldehyde (MDA) and serum enzyme; activate the HO-1/Nrf2 pathway; and increase antioxidant enzymes to protect the liver in CCl_4_-induced chronic liver injury mice. Therefore, MPT-W could be a potentially natural and functional resource contributing to antioxidant, hepatoprotective, and anti-inflammatory effects with potential health benefits.

## 1. Introduction

Chronic liver disease can induce many complications, and it has become a major contributor to high morbidity and mortality rates [1]. Moreover, the World Health Organization has reported that approximately 2.3 million people suffer from liver diseases each year [2]. The liver, the most important metabolic and detoxification organ against various endogenous and exogenous harmful substances, is sensitive to environmental toxins and easily damaged by chemicals such as carbon tetrachloride (CCl_4_) [3]. It is well known that the pathogenesis of liver injury involves a complex interaction of oxidative stress, inflammation, fibrosis, apoptosis, and necrosis [4,5]. CCl_4_, as a representative hepatotoxin, can induce liver injury due to the specific cytochrome P4502E1 (CYP2E1) breaking CCl_4_ into highly reactive trichloromethyl-free radicals (**^·^**CCl_3_^-^ or CCl_3_OO^-^) [6]. A large number of free radicals cannot be eliminated rapidly, causing an imbalance between free radical generation and the antioxidant defense, thereby leading to oxidative stress, which can result in cell damage and death [7]. Antioxidant supplements may suppress oxidative stress-induced injury. Molecular pathogenesis and novel lead drug candidates have been studied and developed for decades, but liver disease treatments continue to be subject to limitations, such as the side effect of chemicals. Therefore, there is motivation to uncover new natural substances against hepatic injury.

*Termitomyces albuminosus*, a well-known symbiotic wild mushroom, grows on termite nests of the African and Asian tropics, and has a symbiotic relationship with the termites [8,9]. The fruiting bodies of *T. albuminosus* are composed of bioactive components, such as polysaccharides, proteins, amino acids, lipids, ergosterol, saponins, cerebrosides, hydrogen peroxide-dependent phenol oxidase, alkaline protease, coumarin and melanin, which are used to strengthen peristaltic ability and in the treatment of some diseases, including intestinal carcinoma, hemorrhoids, hyperlipidemia, hyperglycemia, and antioxidant and antimicrobial diseases [10,11,12,13,14,15,16]. Among these bioactive components, polysaccharides have attracted increasing attention due to their antioxidant, immunomodulating, hepatoprotective, and anti-inflammatory biological activities [11,17,18]. Because the *T. albuminosus* fruiting body cannot be artificially cultivated, *T. albuminosus* mycelia are easily obtained by submerged fermentation, which is a rapid and alternative method. Lu et al. and Zhao et al. have reported that polysaccharides from *T. albuminosus* mycelium possess antioxidant, analgesic, anti-hyperlipidemic and anti-inflammatory effects [11,14]. However, reports on the hepatoprotective effects of mycelium polysaccharides from *T. albuminosus* (MPT) in CCl_4_-induced liver injury mice have been rarely published until the present.

In this study, a major fraction (MPT-W), eluted by deionized water, was extracted from mycelium polysaccharides of *Termitomyces albuminosus* (MPT). The antioxidative, anti-fibrosis and anti-inflammatory activities of MPT-W against CCl_4_-induced chronic liver injury in mice were investigated. Furthermore, the monosaccharide composition, functional groups, configurations and molecular weight (Mw) of MPT-W were also evaluated by gas chromatography-mass spectrometry (GC-MS), Fourier transform infrared spectroscopy (FT-IR) and high-performance gel permeation chromatography (HPGPC).

## 2. Results

### 2.1. Purification

Two peaks were separated by DEAE-52 chromatography (Figure 1A), and a major fraction (MPT-W), eluted by deionized water, was collected. The Sephadex G-100 chromatography of MPT-W had a single, symmetrical peak (Figure 1B), indicating MPT-W was a homogeneous polysaccharide.

### 2.2. FT-IR, Monosaccharide Composition and Mw Analysis

The FT-IR spectrum of MPT-W is displayed in Figure 2A. A strong and broad absorption area at 3427 cm^−1^ manifested the stretching vibration of the hydroxyl group due to intermolecular and intramolecular hydrogen bonds [19]. The presence of the peak at 2920 cm^−1^ was due to the stretching frequency of the C-H bond [20]. In addition, the absorption peaks at 1628 cm^−1^ and 1417 cm^−1^ were the result of the bending vibration of water and the pyranoid ring, respectively [21,22]. The strong characteristic absorptions at 1200–1000 cm^−1^ were due to the vibrations of C-O-C glycosidic bonds. The diagnostic absorption peaks at 830 and 760 cm^−1^ suggested the presence of α and β-type glycosidic linkages [23,24]. Based on these data, it can be concluded that MPT-W is a typical polysaccharide with α- and β-configurations.

As shown in Figure 2B,C, the standard monosaccharides were separated rapidly within 33 min, and their peaks were observed in the order of xylose (Xyl), arabinose (Ara), ribose (Rib), rhamnose (Rha), fucose (Fuc), fructose (Fru), mannose (Man), galactose (Gal), glucose (Glc), and glucuronic acid (GlcA). By comparison with Figure 2B, it was found that MPT-W was made up of Xyl, Fuc, Man, Gal, and Glc with a molar ratio of 0.29:8.67:37.89:35.98:16.60 (Figure 2C).

The profile of MPT-W showed a single and symmetrical peak (Figure 2D), indicating that MPT-W was a homogenous polysaccharide, which was in accord with Sephadex G-100 chromatography. Its number-average molecular weight (Mn), Mw, Z-average molecular weights (Mz) and Mw/Mn were 1.13 × 10^5^ Da, 1.30 × 10^5^ Da, 1.49 × 10^5^ Da and 1.15, respectively (Table 1).

### 2.3. In Vitro Antioxidant Activity of MPT-W

The antioxidant abilities of MPT-W in vitro can be reflected by the conventional DPPH, hydroxyl and superoxide anion radical systems (Table 2). The scavenging activities of MPT-W on DPPH radicals were 0, 11.91% ± 0.84%, 24.87% ± 2.17%, 42.10% ± 1.70%, 63.84% ± 2.80%, 72.01% ± 2.05%, and 76.81% ± 1.31% at concentrations of 0, 200, 400, 600, 800, 1000, 1200 mg/L, respectively. Simultaneously, MPT-W also clearly revealed the scavenging effects against hydroxyl radicals of 0, 19.58% ± 1.98%, 38.10% ± 1.83%, 44.19% ± 2.00%, 54.84% ± 1.91%, 67.18% ± 1.97%, and 74.11% ± 1.52% at the tested concentrations between 0 and 1200 mg/L, respectively. The superoxide anion radical scavenging activity of MPT-W ranging from 0 to 1200 mg/L was shown to be dose dependent and reached 0, 18.88% ± 1.27%, 36.12% ± 0.33%, 52.54% ± 1.170, 57.49% ± 1.84%, 66.16% ± 1.71%, and 72.59% ± 0.58%, respectively. Furthermore, IC_50_ values of MPT-W for scavenging DPPH, hydroxyl and superoxide anion radicals were 638.92 ± 2.81, 613.97 ± 2.79, and 595.26 ± 2.78 mg/L, respectively.

### 2.4. Effect of MPT-W on Liver Injury in CCl_4_-Induced Chronic Liver Injury Mice

The effects of MPT-W on the serum aspartate aminotransferase (AST) and alanine aminotransferase (ALT) activities in CCl_4_-induced chronic liver injury mice are shown in Figure 3A,B. Compared with the normal saline (NS) group, increased activities of serum AST and ALT were found in the administration of the CCl_4_ group (all with *p* = 0.000), indicating that the liver injury model in mice was successfully established. Interestingly, the pretreatment of MPT-W restrained the elevation of serum AST (HMPT-W: *p* = 0.000; LMPT-W: *p =* 0.001) and ALT (HMPT-W: *p* = 0.000; LMPT-W: *p =* 0.001) activities.

To further verify liver injury status, hemotoxylin and eosin (H&E) staining was applied to evaluate hepatic histology (Figure 3C). Liver section from the NS group showed an ordered manner liver cell with normal cellular morphology, abundant cytoplasm, intact nucleus, and well-defined cell borders. Inversely, long-term injection of CCl_4_ caused hepatic cell disorders, non-distinct cellular boundaries, nuclear loss and a large area of hepatocyte necrosis in the model control (MC) group. However, MPT-W pre-treatment attenuated CCl_4_-induced liver damage.

### 2.5. Effect of MPT-W on Antioxidant Status in CCl_4_-Induced Chronic Liver Injury Mice

CYP2E1 can convert CC1_4_ into toxic metabolites such as **^·^**CCl_3_^−^ or CCl_3_OO^−^, which can cause cell damage and its death. Inhibiting CYP2E1 activity can prevent the metabolism of CC1_4_ into free radicals, and thereby plays a role in protecting the liver. To clarify the preventive mechanism of MPT-W on oxidative stress, the hepatic CYP2E1 was evaluated (Figure 4A,B). The CYP2E1 level and its mRNA expression in the MC group had an obvious increase, as compared to the NS group (all with *p*
*=* 0.000), which provided evidence of CCl_4_-induced hepatocyte damage. MPT-W pre-treatment obviously reduced the CYP2E1 level (HMPT-W: *p* = 0.001; LMPT-W: *p =* 0.007) and its mRNA expression (all with *p =* 0.01), compared to the MC group.

As shown in Figure 4C–E, remarkable decreases in superoxide dismutase (SOD, *p* = 0.000), GSH peroxide (GSH-Px, *p* = 0.001) and catalase (CAT, *p* = 0.000) activities were observed in the MC group, when compared with the NS group (*p* < 0.05), indicating that the administration of CCl_4_ had damaged the hepatic antioxidative defense system. The amount of hepatic SOD, GSH-Px, and CAT activities in the MC group decreased by 49.90%, 50.07% and 40.43%, respectively, when compared with those of the NS group (138.95 ± 8.17 U/mg prot, 68.81 ± 5.42 U/mg prot, 138.19 ± 6.68 U/mg prot). However, the activities of SOD (HMPT-W: *p* = 0.000; LMPT-W: *p =* 0.001), GSH-Px (HMPT-W: *p* = 0.000; LMPT-W: *p =* 0.001) and CAT (HMPT-W: *p* = 0.001; LMPT-W: *p =* 0.003) were significantly enhanced by the administration of MPT-W.

The level of hepatic malondialdehyde (MDA) in the MC group (10.06 ± 0.25 mol/g prot) was clearly raised compared with that in the NS group (2.45 ± 0.14 mol/g prot) (*p* = 0.000, Figure 4F), indicating that membrane lipid peroxidation had been initiated. As expected, MPT-W distinctly suppressed this abnormal change (all with *p* = 0.000).

To further investigate the scavenging ability of MPT-W against free radicals, reactive oxygen species (ROS) level was determined by dihydroethidium label (Figure 5A,B). The ROS level in the MC group was higher than that of the NS group (*p* = 0.001), indicating that the injection of CCl_4_ can induce the generation of mass ROS. However, the abnormal change of ROS level was dramatically improved by the treatment with MPT-W (HMPT-W: *p* = 0.01; LMPT-W: *p =* 0.037).

As exhibited in Figure 5C and 5D, CCl_4_ injection observably upregulated the mRNA expressions of HO-1 (*p* = 0.005) and Nrf2 (*p* = 0.002), as compared to the NS group, showing that CCl_4_ can activate Nrf2 and HO-1. Interestingly, the supplementation of MPT-W further increased mRNA expression of HO-1 (HMPT-W: *p* = 0.001; LMPT-W: *p =* 0.008) and Nrf2 (HMPT-W: *p* = 0.003; LMPT-W: *p =* 0.004).

### 2.6. Effect of MPT-W on Fibrosis in CCl_4_-Induced Chronic Liver Injury Mice

As displayed in Figure 6A,B, sections from the MC group revealed a greater collagen volume fraction compared to that of the NS group (*p* = 0.000), showing that severe fibrosis had occurred in liver tissue due to long-term injection of CCl_4_. In comparison with the MC group, HMPT-W (*p* = 0.000) and LMPT-W (*p* = 0.001) significantly decreased the collagen volume fraction by 68.58% and 31.81%, respectively, indicating that MPT-W showed antifibrogenic effect.

To provide more evidence about the antifibrogenic effect of MPT-W, TGFβ1, and Smad3 mRNA expressions, as well as TGFβ1 and Smad3 protein expressions, were evaluated (Figure 6C–G). The expression quantities of TGFβ1 and Smad3 mRNAs, as well as TGFβ1 and Smad3 proteins, were obviously upregulated (all with *p* = 0.000) in the MC group, as compared with the NS group. However, various MPT-W-treated groups significantly inhibited these growth trends when compared to those of the MC group (all with *p* = 0.000). These results showed that the protective effect of MPT-W on liver damage in CCl_4_-induced chronic liver injury mice might be related to regulation of the TGF-β1/Smad3 signaling pathway.

### 2.7. Effect of MPT-W on Hepatic Inflammatory Response in CCl_4_-Induced Chronic Liver Injury Mice

As shown in Figure 7, the mRNA expressions of inflammatory cytokines (tumor necrosis factor: TNF-α, interleukin-6: IL-6 and interleukin-1β: IL-1β) in liver were significantly upregulated in the MC group, compared with those in the NS group (all with *p* = 0.000), indicating that an inflammation response was triggered by the injection of CCl_4_. However, the supplementation of MPT-W obviously suppressed the upregulation of inflammatory cytokines, when compared to those of the MC group (all with *p* = 0.000).

To further understand the anti-inflammatory mechanism of MPT-W, we investigated the protein expressions of IκBα, p-IKK and p-NF-κB p65 in the NF-κB signaling pathway by immunoblotting (Figure 8). It was observed that the protein expressions of p-IKK and p-NF-κB p65 were remarkably increased in the MC group (all with *p* = 0.000), while the protein expression of IκBα was obviously decreased, compared with those in the NS group (*p* = 0.000), demonstrating that CCl_4_ had activated the NF-κB signaling pathway. The administration of MPT-W reduced the protein expressions of p-IKK (HMPT-W: *p* = 0.001; LMPT-W: *p =* 0.004) and p-NF-κB p65 (HMPT-W: *p* = 0.000; LMPT-W: *p =* 0.000), and enhanced IκBα (HMPT-W: *p* = 0.000; LMPT-W: *p =* 0.000) protein expression.

To provide more evidence to support the Western blot results of IκBα, p-IKK and p-NF-κB p65, immunohistochemical staining of IκBα, p-IKK and p-NF-κB p65 in the liver was undertaken (Figure 9). Compared to the NS group, the mean densities of p-IKK (*p* = 0.000) and p-NF-κB p65 (*p* = 0.010) in the MC group were significantly increased, while IκBα mean density was markedly decreased (*p* = 0.001). Interestingly, MPT-W supplementation obviously improved these abnormal changes. Immunohistochemistry analysis result was in accordance with that of Western blot.

## 3. Discussion

The present study was designed to explore the protective effect of MPT-W against CCl_4_-induced chronic liver injury. Furthermore, the underlying mechanisms of MPT-W on oxidative stress, inflammatory reaction and fibrosis signaling pathways were explained. Results showed that MPT-W can attenuate CCl_4_-induced liver injury by enhancing the antioxidant and anti-inflammatory abilities, as well as reducing fibrosis.

The accumulated literature has reported that the oxidative stress is responsible for many human diseases, and liver diseases are still a serious global health issue [25,26]. Hydroxyl radicals with the strongest chemical activity among various ROS can contribute to the damage of biomolecules such as protein and nucleic acid [27]. Superoxide anion radicals, relatively weak oxidants and the most common free radicals produced in vivo, are one of the precursors of O_2_ and hydroxyl radicals, which can cause tissue damage [28]. Moreover, Li, Chen, Wang, Tian, and Zhang have reported that superfluous superoxide anion radicals are considered to be the onset of ROS gathering in cells thereby leading to the imbalance of redox and the consequences of correlative detrimental physiology [29]. The DPPH radical is a stable free radical that can combine electrons/H to form a stable diamagnetic molecule (DPPH-H), and the evaluation of scavenging activity on DPPH is a rapid and efficient method for assaying antioxidant activity [30]. Hence, it would be of great significance to explore a natural compound with good scavenging capacities on radicals of hydroxyl, DPPH, and superoxide anions for treating and preventing ROS-induced diseases. In the assessment of the antioxidant ability of MPT-W, strong scavenging activities against hydroxyl, DPPH and superoxide anion radicals were revealed. In addition, these results were further verified in an in vivo mouse model suffering from oxidative stress induced by CCl_4_.

Serum AST and ALT are used as marker enzymes for detecting liver injury, and their elevations in serum show that the permeability and structural integrity of hepatocyte are damaged, causing AST and ALT leakage into the serum [31]. In our work, serum AST and ALT activities in the MC group were obviously elevated by the injection of CCl_4_. Moreover, Rocha et al. also have reported that CCl_4_ can lead to liver damage with increases of AST and ALT [1]. The administration of MPT-W could suppress the change of AST and ALT activities in CCl_4_-induced chronic liver injury mice, indicating that MPT-W could enhance cytomembrane stability and structural integrity to treat and prevent CCl_4_-induced chronic liver injury, which was in accord with histopathological analysis. Liu, Zheng, Su, Wang, and Li reported that *Agaricus bisporus* polysaccharides showed significant hepatoprotective activity in CCl_4_-induced chronic liver injury mice [32].

Many studies have shown that oxidative stress, a major mechanism, was concerned with CCl_4_ toxicity [30]. CYP2E1, one of the isoforms of the CYP450 system, is responsible for the metabolization of CCl_4_ into trichloromethyl free radicals, which can attack lipids, membrane proteins and thiols to cause liver oxidative stress injury [33]. However, these free radicals could be effectively scavenged by some important antioxidant enzymes, such as SOD, GSH-Px and CAT, which disable free radicals-activated lipid peroxidation [34]. MDA, an index for estimating lipid peroxidation, is a metabolism of the lipid peroxidation-generated end product and is capable of destroying the cell [35]. In the present study, we have proved that MPT-W could distinctly enhance the SOD, GSH-Px and CAT activities, and reduce CYP2E1 and MDA levels in CCl_4_-poisoned mice, indicating that MPT-W had potentially treated and prevented CCl_4_-induced liver injury by increasing antioxidant ability and decreasing CCl_4_ metabolism and lipid peroxidation. This finding was supported by ROS level evaluation.

Nrf2, a key transcriptional factor, plays an important role in regulating the antioxidant defense, and can regulate genes encoding HO-1 and other antioxidant enzymes [36]. Peng, Dai, Liu, Li, and Qiu have reported that CCl_4_ is able to activate hepatic Nrf2 [37]. In our work, CCl_4_ injection remarkably increased the mRNA expressions of Nrf2. Interestingly, their mRNA expressions were further enhanced by treatment with MPT-W, which implied a molecular basis for MPT-W to irritate the antioxidant and phase II detoxifying enzymes. These results also supported the finding that MPT-W could increase antioxidant activities.

TGF-β1, a member of the pleiotropic cytokine family, plays a leading role in the development of liver fibrosis and improves extracellular matrix production, and can activate Smad3 to accelerate fibrosis progression [32,38]. Some reports have shown that the activation of the TGF-β1/Smad3 pathway was related to liver injury in CCl_4_-induced chronic liver injury mice [37]. In our work, the mRNA expressions of TGF-β1 and Smad3, as well as the protein expressions of TGF-β1 and Smad3, were significantly increased in the MC group compared to the NS group, and markedly decreased in MPT-W-treated groups. These results showed MPT-W could inhibit the TGF-β1/Smad3 signaling pathway for preventing CCl_4_-induced liver injury. Masson staining also indicated that MPT-W decreased fibrosis progression.

Mounting evidence has showed that inflammation plays an important pathomechanism in hepatic injury and can accelerate the progression of liver damage [39]. Hence, inhibiting the secretion of pro-inflammatory cytokines is an effective measure to attenuate CCl_4_-induced liver injury. In our work, the mRNA expressions of TNF-α, IL-6 and IL-1β were visibly upregulated in the MC group but were significantly downregulated in the MPT-W-treated mice, indicating that MPT-W improved liver injury, possibly by repressing the inflammatory cytokine expressions. To learn more about the underlying mechanism of MPT-W against CCl_4_-induced inflammatory reaction, we also assessed the activation of the NF-κB signal pathway, which plays a pivotal role in expressions of many pro-inflammatory cytokines. The NF-κB dimer (p65/p50), released by IκBα phosphorylated via activated IKK, ubiquitinated and degraded, was translocated into the nucleus to provoke the target gene transcriptions [40,41]. Our results showed that CCl_4_ significantly stimulated the NF-κB pathway by increasing p-IκBα, IKK and p-NF-κB p65. However, the supplement of MPT-W dramatically reduced the NF-κB activation.

## 4. Materials and methods

### 4.1. Materials

The *T. albuminosus* strain was obtained by our laboratory and used in this work. The reagents for analyzing SOD, GSH-Px, CAT, and MDA were purchased from Nanjing Jiancheng Bioengineering Co., Ltd. (Nanjing, China). The kit of CYP2E1 was obtained from Jiangsu Meibiao Biological Technology Co., Ltd. (Jiangsu, China). Monosaccharide standards were obtained from Sigma Chemicals Co. Ltd., (St. Louis, MS, USA). All other chemicals of analytical grade used in this study were obtained from local chemical suppliers.

### 4.2. Extraction and Purification of MPT

The dried mycelium (100 g) of *T. albuminosus* was crushed into powder, which was mixed with water (2 L) at 90 °C for 3 h. After centrifugation at 3000 rpm for 5 min, the collected and concentrated supernatant was mixed with 3 volumes of 95% *v*/*v* ethanol at 4 °C overnight to offer precipitate. The deproteinated sediments obtained by the Sevage method [42] were dialyzed against deionized water and lyophilized to give MPT powder.

MPT powder was distilled in water and filtrated by 0.22-µm filter membrane. Subsequently, MPT solution was used in a DEAE-cellulose column (1.8 × 30 cm), and eluted with gradient NaCl solutions of 0.0, 0.1, 0.3, 0.5 and 1 M (4 mL/tube). The collected fractions were dialyzed against distilled water, and lyophilized to powder.

To improve the main fraction purity, the main fraction was fractionated by size-exclusion chromatography on a Sephadex G-100 column with distilled water (4 mL/tube), collected, dialyzed and lyophilized to powder, for use in further studies.

### 4.3. Monosaccharide Compositions Analysis

The monosaccharide compositions were analyzed by GC-MS (Agilent5975c, Palo alto, CA, USA). MPT-W (2.0 mg) was hydrolyzed with trifluoroacetic acid (5.0 mL, 2.0 M) in N_2_ atmosphere for 4 h at 120 °C. The absolute methanol was added to the above reaction mixture, rotated and steamed repeatedly to remove trifluoroacetic acid, and then dissolved in deionized water (2 mL). MPT-W hydrolysate (100 µL) was mixed with deuterium-labeled succinic acid (10 µL, 1.5 mg/mL), and lyophilized to powder, which was added to methoxammonium hydrochloride/pyridine solution (50 μL, 20 mg/mL) in a water bath pot at 40 °C for 80 min. After termination of the reaction, *N*-methyl-*N*-(trimethylsilyl) trifluoroacetamide (80 μL) was mixed with the reaction solution in water bath pot at 40 °C for 80 min. After centrifugation (12,000 rpm, 10 min), the supernatant was placed for 2 h at room temperature, and then analyzed by GC-MS. Monosaccharide compositions were determined using the standard curves of Rha, Ara, Xyl, Man, Gal, Glc, GlcA, Rib, Fru and Fuc.

### 4.4. FT-IR Spectroscopy Analysis

FT-IR spectroscopy was recorded by a 6700 Nicolet Fourier transform-infrared spectrophotometer (Thermo Co., Madison, WI, USA) within the range from 4000 to 400 cm^−1^, using the KBr disc method to prepare the specimen.

### 4.5. Mw Determination

The molecular weights were estimated by HPGPC using an HPLC system (Agilent 1260, Agilent Technologies, Palo alto, CA, USA). MPT-W (100 µL) was injected into a Shodex SB-806HQ column (8 mm × 300 mm) with the mobile phase (deionized water) at a flow rate of 1.0 mL/min. A series of standard dextrans was used to make the calibration curve, and the molecular weight was analyzed by Agilent GPC software.

### 4.6. In Vitro Antioxidant Activity of MPT-W

The scavenging activity of MPT-W against DPPH radicals was measured as previously described [43]. Briefly, DPPH in methanol solution (0.6 mL, 0.004%, *w*/*v*) was mixed with MPT-W preparative solution (0.2 mL) at 0.0, 0.2, 0.4, 0.6, 0.8, 1.0 and 1.2 g/L. After the mixture was kept in the dark for 30 min, the *OD*
_517 nm_ values were determined by a visible spectrometer using deionized water as the blank and vitamin C (Vc) as a positive control.

The scavenging activity of MPT-W on hydroxyl radicals was evaluated according to our previous procedure [44]. MPT-W at various concentrations (500 µL, 0.0–1.2 g/L) was added to a mixture solution of 2-desoxyribose in phosphate buffer (100 µL, 28 mM, pH 7.4), ferric trichloride solution (100 µL, 200 µM), ethylenediamine tetraacetic acid disod (100 µL, 1.04 mM), hydrogen peroxide solution (100 µL, 1 mM) and ascorbic acid solution (100 µL, 1 mM). The reaction mixture was incubated for 1 h at 37 °C and *OD _5_*_32 nm_ values were measured by a visible spectrometer using deionized water as the blank and Vc as a positive control.

The scavenging superoxide anion radical ability of MPT-W was determined using the method reported by Nishimiki, Rao, and Yagi [45]. In the phenazine methosulfate-nicotinamide adenine dinucleotide system, sodium phosphate buffer (3.0 mL, 100 mM, pH 7.4), various concentrations of MPT-W (1.0 mL), nitroblue tetrazolium solution (1.0 mL, 150 µM), nicotinamide adenine dinucleotide solution (1.0 mL, 468 µM) and phenazine methosulfate (1.0 mL, 60 µM) were mixed, incubated at 25 °C for 5 min, and then *OD*_560 nm_ values were read by a visible spectrometer using deionized water as the blank and Vc as a positive control.

The scavenging rate of MPT-W against DPPH, hydroxyl and superoxide anion radicals was calculated by the following equation:$$Scavenging\;rate\;\left(\%\right)\;=\frac{\left(\mathrm{Aa}-\mathrm{Ab}\right)}{Aa}\times 100$$
where A_a_ is absorbance of the blank only without sample and A_b_ is absorbance of MPT-W or Vc.

The IC_50_ values (μg/mL) of scavenging DPPH, hydroxyl or superoxide anion radicals were defined as the effective concentrations of the sample at which the radicals were inhibited by 50%.

### 4.7. Animal Experiment

Fifty Kunming male mice (20 ± 2 g) were provided by Jinan pengyue experimental animal breeding Ltd. Co. (Jinan, China). During the experiment, the mice were given free drinking water and a standard diet in standard laboratory conditions (25 ± 2 °C, relative humidity 50% ± 10%, 12 h light/dark cycles). All experiments were conducted in accordance with Institutional Animal Care, and approved by the Committee of Shandong Agricultural University. After adaptive feeding for seven days, all mice were randomly divided into 5 groups (10 each group), namely the NS group, MC group, positive control (PC) group, HMPT-W group and LMPT-W group. During the experimental procedure, all groups except the NS group were injected with mixture (0.5%, 10 mL/kg CCl_4_ + peanut oil) into enterocoelia (two injections a week), and the NS group were injected with equal volume saline. After 8 weeks, the HMPT-W and LMPT-W groups received 400 mg/kg and 200 mg/kg MPT-W, respectively, the PC group was treated with 200 mg bifendate/kg, and the NS and MC groups were administrated with normal saline. The whole experiment lasted 14 weeks. After the last gavage for 24 h, all mice were sacrificed by anesthesia, and blood and liver samples were collected.

### 4.8. Enzyme Activities in Serum Assessment

The blood samples were centrifuged (5000 rpm, 4 °C, 10 min), and the serum was collected. The activities of serum AST and ALT were measured in serum supernatants using an automatic biochemical analyzer (Beckman Coulter, Fullerton, CA, USA).

### 4.9. Biochemical Indices Evaluation in Liver Homogenates

The liver homogenates, obtained from fresh liver, were homogenized using 0.2 M phosphate buffer solutions (pH 7.4) and centrifuged at 3000 rpm for 10 min, and used for evaluating SOD, GSH-Px, CAT, CYP2E1 and MDA by commercial kits according to the instructions.

### 4.10. Western Blotting

Liver tissue (0.1 g) was homogenized using 990 µL ice-cold RIPA lysis buffer, 10 µL protease inhibitor cocktail and 10 µL phosphatase inhibitor cocktail by glass homogenizer in ice water for 20 min. After centrifugation at 10,000 rpm for 10 min, the supernatant was mixed with loading buffer (5 ×) and boiled for 10 min. The protein concentration was measured using the Bradford protein assay kit. Equal amounts of protein from each sample were separated using 15% SDS-PAGE gel electrophoresis and transferred to PVDF membranes, which were then blocked in 5% (*w*/*v*) non-fat powdered milk for 2 h. Primary rabbit antibodies against IκBα and p-IKK (1:1000) (Cell Signaling Technology, Inc., Boston, MA, USA), p-NF-κB p65 (Absin biotechnology Co., Ltd., Shanghai, China), HO-1, Nrf2, TGF-β1 and Smad3 (1:500) (Wanleibio Co., Ltd., Shenyang, China), and GAPDH (1:1000) (Zhixian Biological Co., Ltd., Hangzhou, China) were employed. Goat anti-rabbit IgG/HRP antibodies (1:5000) (Beijing Solarbio Science & Technology Co., Ltd., Beijing, China) were employed as the secondary antibodies.

### 4.11. Quantitative Reverse Transcription-Polymerase Chain Reaction (qRT-PCR) Analysis

Total RNA of mice liver was extracted using Trizol reagent. The purity of total RNA was estimated by OD_260_/OD_280_, and RNA of all samples was in the range from 1.9 to 2.1. cDNA synthesis was performed using the TransScript^®^ All-in-One First-Strand cDNA Synthesis SuperMix kit (TransGen Biotech Co., Ltd., Beijing, China). RT-PCR was performed using ChamQ^TM^ Universal SYBR^®^ qPCR Master Mix. Primer was synthesized by Beijing ruiboxingke biotechnology Co., Ltd., Beijing, China. The PCR primer sequences used were as follows: IL-6 forward: 5′-TAC CAC TCC CAA CAG ACC TG-3′; IL-6 reverse: 5′-GGT ACT CCA GAA ACC AGA GG-3′; TNF-α forward: 5′-CAC CAT GAG CAC AGA AAG CA-3′; TNF-α reverse: 5′-TAG ACA GAA GAG CGT GGT GG-3′; IL-1β forward: 5′-ACT CAT TGT GGC TGT GGA GA-3′; IL-β reverse: 5′-TTG TTC ATC TCG GAG CC TGT-3′; CYP2E1 forward: 5′-CCA CCA GCA CAA CTC TGA GAT A-3′; CYP2E1 reverse: 5′-CCC AAT AAC CCT GTC AAT TTC TT-3′; TGF-β1 forward: 5′-GAT TGT TGC CAT CAA CGA CC-3′; TGF-β1 reverse: 5′-GTG CAG GAT GCA TTG CTG AC-3′; Smad3 forward: 5′-CCA GCA CAC AAT AAC TTG GA-3′; Smad3 reverse: 5′-AGA CAC ACT GGA ACA GCG GA-3′; β-actin forward: 5′-ATT CGT TGC CGG TCC ACA CCC-3′; β-actin reverse: 5′-GCT TTG CAC ATG CCG GAG CC-3′.

### 4.12. Histopathological Analysis

Livers, fixed in 4% paraformaldehyde, were embedded in paraffin, cut into 4-µm slices, stained with H&E or Masson staining, and finally observed and photographed under a microscope (200× magnification).

### 4.13. Immunohistochemistry Staining

Paraffin-embedded sections of the mice liver were used to detect p-NF-κB p65, IκBα and p-IKK using anti-p-NF-κB p65 (1:250 dilution, Absin biotechnology Co., Ltd., Shanghai, China), and anti-IκBα and anti-IKK (1:100 dilution, Cell Signaling Technology, Inc., Boston, MA, USA), and the secondary antibodies (1:500 dilution of HRP-goat anti-rabbit, Absin biotechnology Co., Ltd., Shanghai, China). Briefly, paraffin-embedded section was performed by dewaxing, antigen retrieval, 3% hydrogen peroxide incubation, goat-serum incubation, antibody incubation, secondary antibody incubation, DAB coloration, hematoxylin counterstaining, dehydration, transparency treatment, mounting, and microscopic examination.

### 4.14. ROS Level Assay

Frozen sections of fresh tissue were rewarmed, dyed by dihydroethidium, dyed for nucleus by DAPI, mounted by anti-fluorescence quenching sealing tablets, and observed by fluorescence microscope [46]. ROS staining effect is red, and the amount and change of ROS content can be determined according to the strength of red fluorescence in cells.

### 4.15. Statistical analysis

All results were expressed as the mean ± SD from 3 independent experiments. Differences among groups were analyzed using one-way ANOVA with a post hoc Duncan’s multiple range tests using SPSS software. *p*-values less than 0.05 between experimental groups were considered significant.

## 5. Conclusions

The MPT-W from *T. albuminosus* had potential antioxidant, anti-fibrosis and anti-inflammatory activities to prevent liver injury in CCl_4_-induced mice. Based on the present results, the MPT-W might be a potentially natural and functional resource with potential health benefit. To be further applied to clinical practice, its pesticide effect and toxicity need to be further studied, and data quality needs to be monitored. In addition, MPT-W yield should be optimized and its structure should be further analyzed.

## Abbreviation

AraArabinoseCCl_4_Carbon tetrachlorideCATCatalaseCYP2E1Cytochrome P4502E1FT-IRFourier-transform infrared spectroscopyFruFructoseFucFucoseGalGalactoseGC-MSGas chromatography-mass spectrometryGlcGlucoseGlcAGlucuronic acidGSH-PxGSH peroxideHPGPChigh performance gel permeation chromatographyIL-1βInterleukin-1βIL-6Interleukin-6MDAMalonaldehydeManMannoseMCmodel controlMwMolecular weightMPTMycelium polysaccharides from *T. albuminosus*NSNormal salineMnNumber-average molecular weightsPCpositive controlROSReactive oxygen speciesRhaRhamnoseRiRiboseSODSuperoxide dismutaseTNF-αTumor necrosis factorXylXyloseMzZ-average molecular weightsHMPT-W400 mg/kg MPT-WLMPT-W100 mg/kg MPT-W

## Figures and Tables

**Figure 1 ijms-20-04872-f001:**
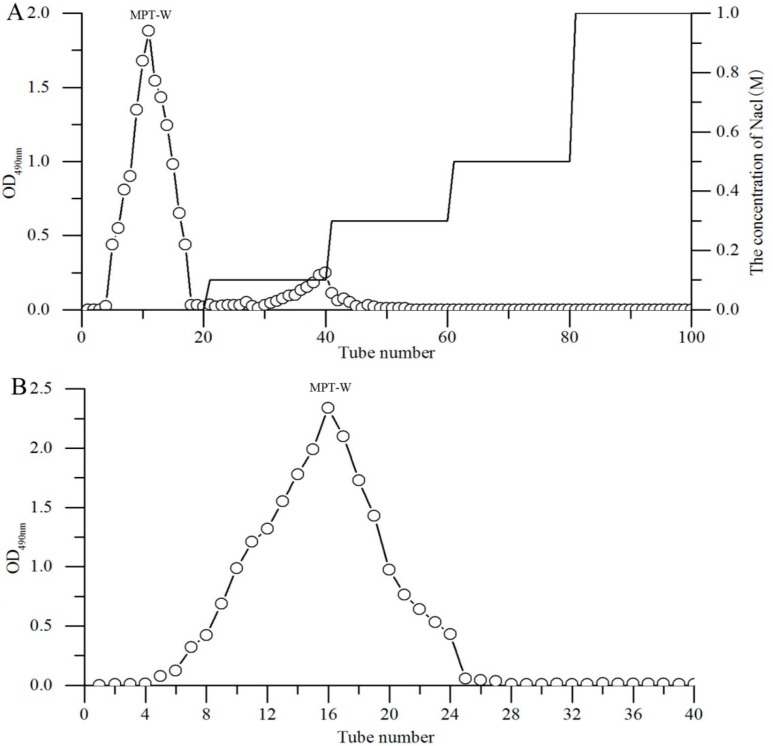
Elution profile of *Termitomyces albuminosus* (MPT). (**A**) DEAE-52 cellulose column chromatography; (**B**) Sephadex G-100 column chromatography.

**Figure 2 ijms-20-04872-f002:**
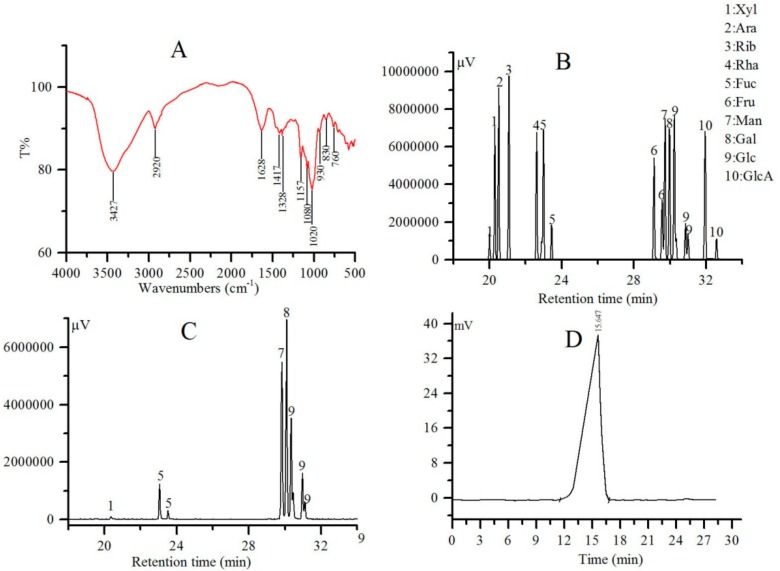
FT-IR, monosaccharide composition and HPGPC analysis. (**A**) FT-IR; (**B**) GC-MS of standard samples; (**C**) GC-MS of MPT-W; (**D**) HPGPC. FT-IR: Fourier transform infrared spectroscopy, HPGPC: high performance gel permeation chromatography, GC-MS: gas chromatography-mass spectrometry.

**Figure 3 ijms-20-04872-f003:**
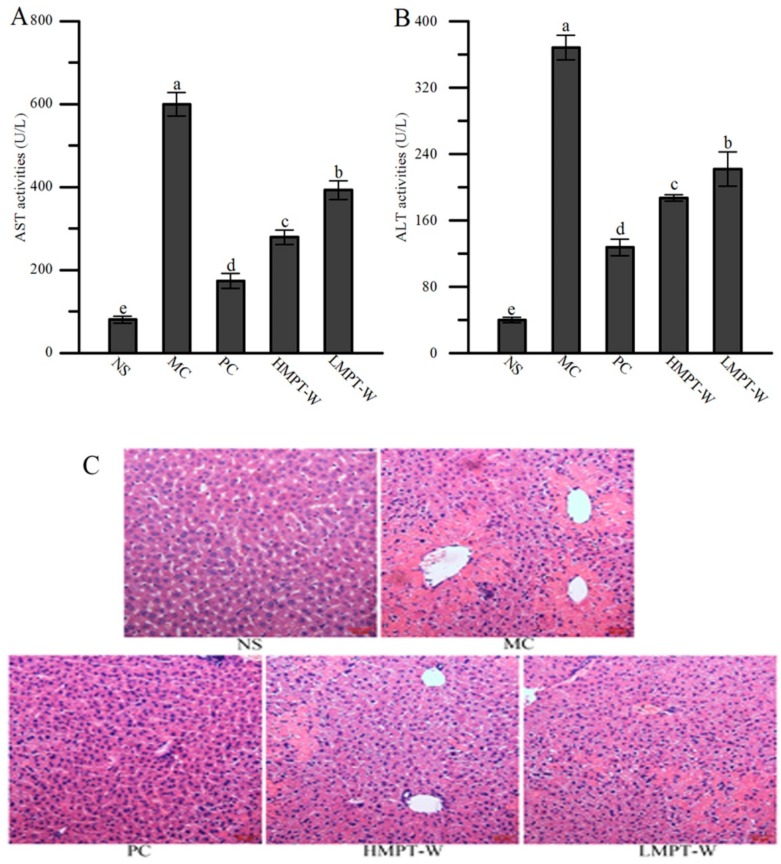
Effect of MPT-W on liver injury in CCl_4_-induced chronic liver injury mice. (**A**) aspartate aminotransferase (AST) and (**B**) alanine aminotransferase (ALT) in serum; (**C**) Histopathological images of H&E stained liver sections from the mice of NS, MC, PC, HMPT-W, and LMPT-W groups, magnification 200×. The values are reported as means ± SD. Bars with different letters (a, b, c, d, e) are significantly different (*p* < 0.05). NS: normal saline, MC: model control, PC: positive control, HMPT-W: 400 mg/kg MPT-W, LMPT-W: 200 mg/kg MPT-W, AST: aspartate aminotransferase, ALT: alanine aminotransferase.

**Figure 4 ijms-20-04872-f004:**
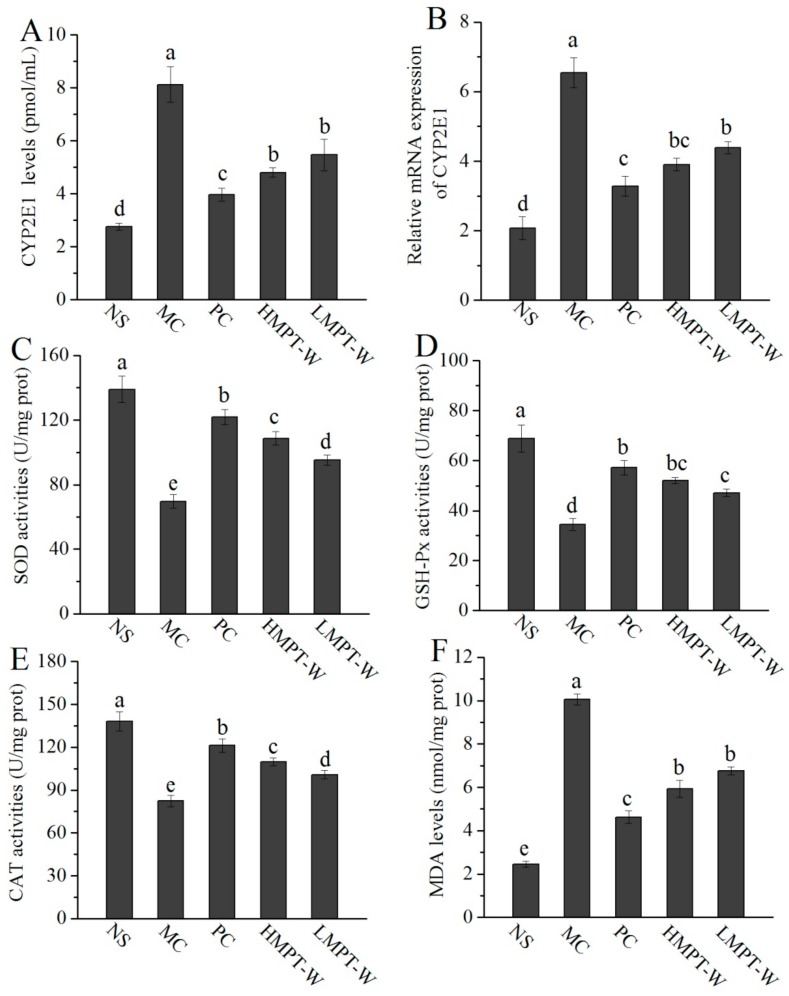
Effect of MPT-W on hepatic CYP2E1, antioxidant enzymes and MDA in CCl_4_-induced chronic liver injury mice. (**A**) and (**B**) CYP2E1; (**C**) SOD; (**D**) GSH-Px; (**E**) CAT; (**F**) MDA. The values are reported as means ± SD. Bars with different letters (a, b, c, d, e) are significantly different (*p* < 0.05). CYP2E1: cytochrome P4502E1, MDA: malondialdehyde, SOD: superoxide dismutase, GSH-Px: GSH peroxide, CAT: catalase.

**Figure 5 ijms-20-04872-f005:**
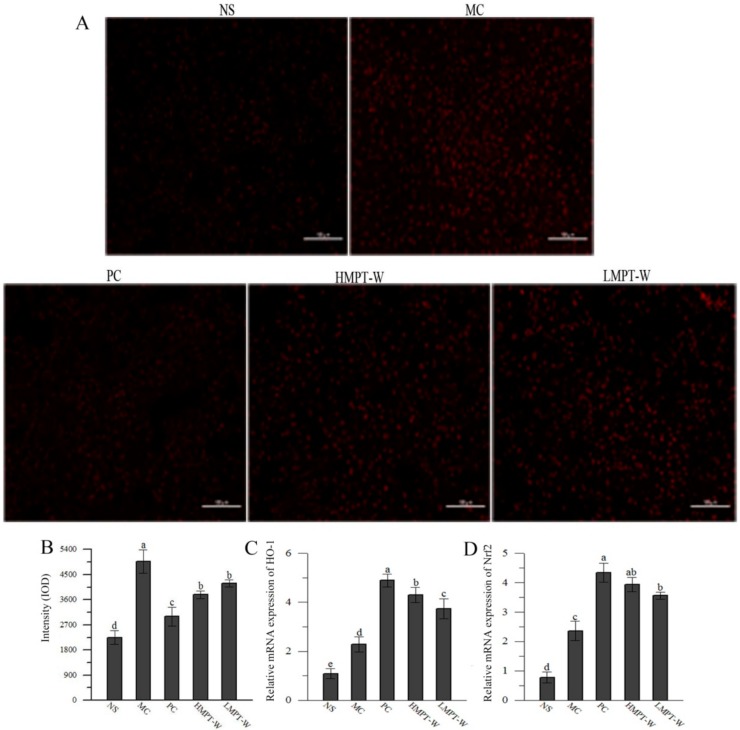
Effect of MPT-W on hepatic ROS, HO-1, and Nrf2 in CCl_4_-induced chronic liver injury mice. (**A**) ROS fluorescence labeling images of liver sections from NS, MC, PC, HMPT-W and LMPT-W treated mice, magnification 200×; (**B**) ROS quantification; (**C**) HO-1; (**D**) Nrf2. The values are reported as means ± SD. Bars with different letters (a, b, c, d, e) are significantly different (*p* < 0.05). ROS: reactive oxygen species, HO-1: heme oxygenase-1, Nrf2: nuclear factor erythroid-2-related factor 2.

**Figure 6 ijms-20-04872-f006:**
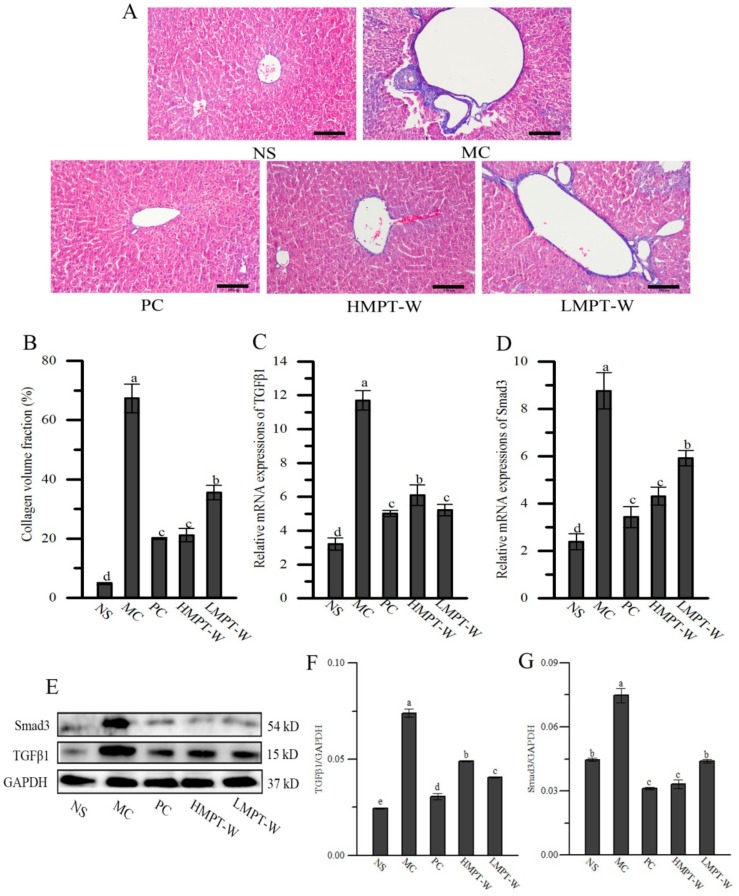
Effect of MPT-W on hepatic fibrosis in CCl_4_-induced chronic liver injury mice. (**A**) Masson staining images of liver sections from NS, MC, PC, HMPT-W and LMPT-W treated mice, magnification 200×; (**B**) collagen volume fraction; (**C**) TGFβ1 mRNA; (**D**) Smad3 mRNA; (**E**) blotting of TGFβ1 and Smad3; (**F**) TGFβ1/GAPDH; (**G**) Smad3/GAPDH. The values are reported as means ± SD. Bars with different letters (a, b, c, d, e) are significantly different (*p* < 0.05). NS: normal saline, MC: model control, PC: positive control, HMPT-W: 400 mg/kg MPT-W, LMPT-W: 200 mg/kg MPT-W, TGFβ1: transforming growth factor beta 1, Smad3: drosophila mothers against decapentaplegic protein-3, GAPDH: phosphoglyceraldehyde dehydrogenase.

**Figure 7 ijms-20-04872-f007:**
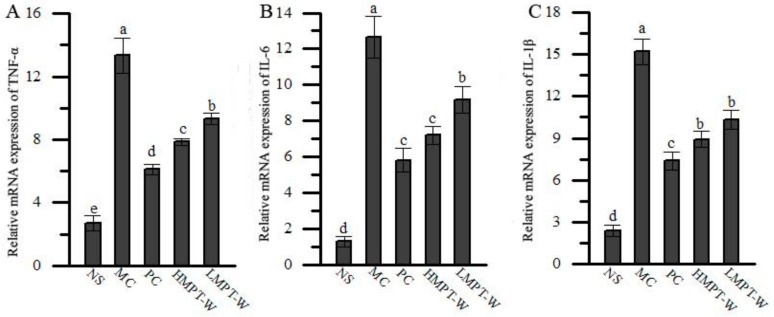
Effect of MPT-W on hepatic inflammatory cytokines in CCl_4_-induced chronic liver injury mice. (**A**) TNF-α; (**B**) IL-6; (**C**) IL-1β. The values are reported as means ± SD. Bars with different letters (a, b, c, d, e) are significantly different (*p* < 0.05). TNF-α: tumor necrosis factor, IL-6: Interleukin-6, IL-1β: Interleukin-1β.

**Figure 8 ijms-20-04872-f008:**
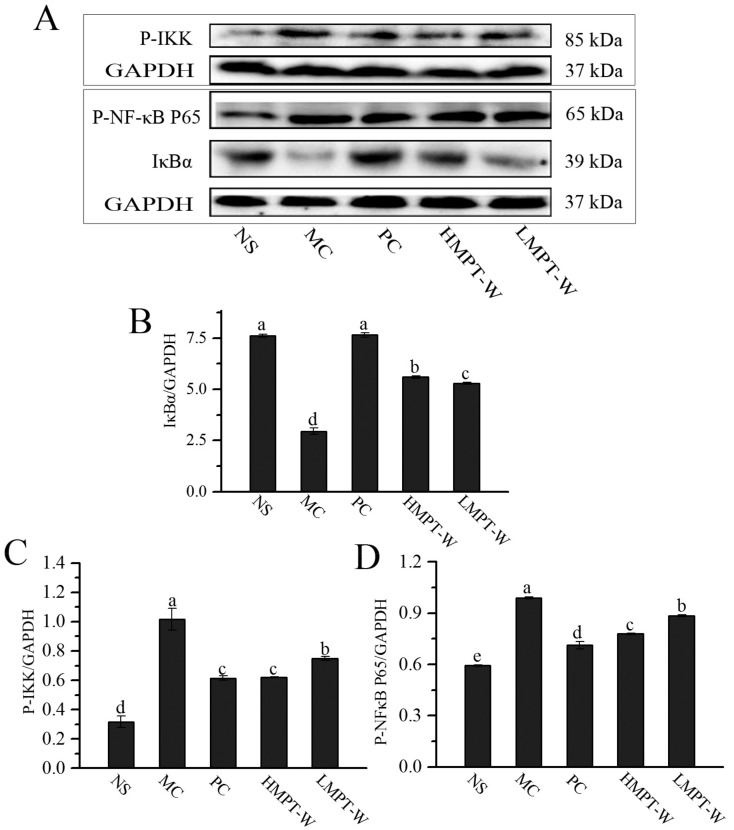
Effect of MPT-W on NF-κB signaling pathway by immunoblotting in CCl_4_-induced chronic liver injury mice. (**A**) Western blot image; (**B**) IκBα/GAPDH; (**C**) p-IKK/GAPDH; (**D**) p-NF-κB p65/GAPDH. The values are reported as means ± SD. Bars with different letters (a, b, c, d, e) are significantly different (*p* < 0.05). p-NF-κB: phospho-nuclear factor-kappa B, IκBα: NF-kappa-B inhibitor alpha, p-IKK: phospho-inhibitor of nuclear factor kappa-B kinase, GAPDH: phosphoglyceraldehyde dehydrogenase.

**Figure 9 ijms-20-04872-f009:**
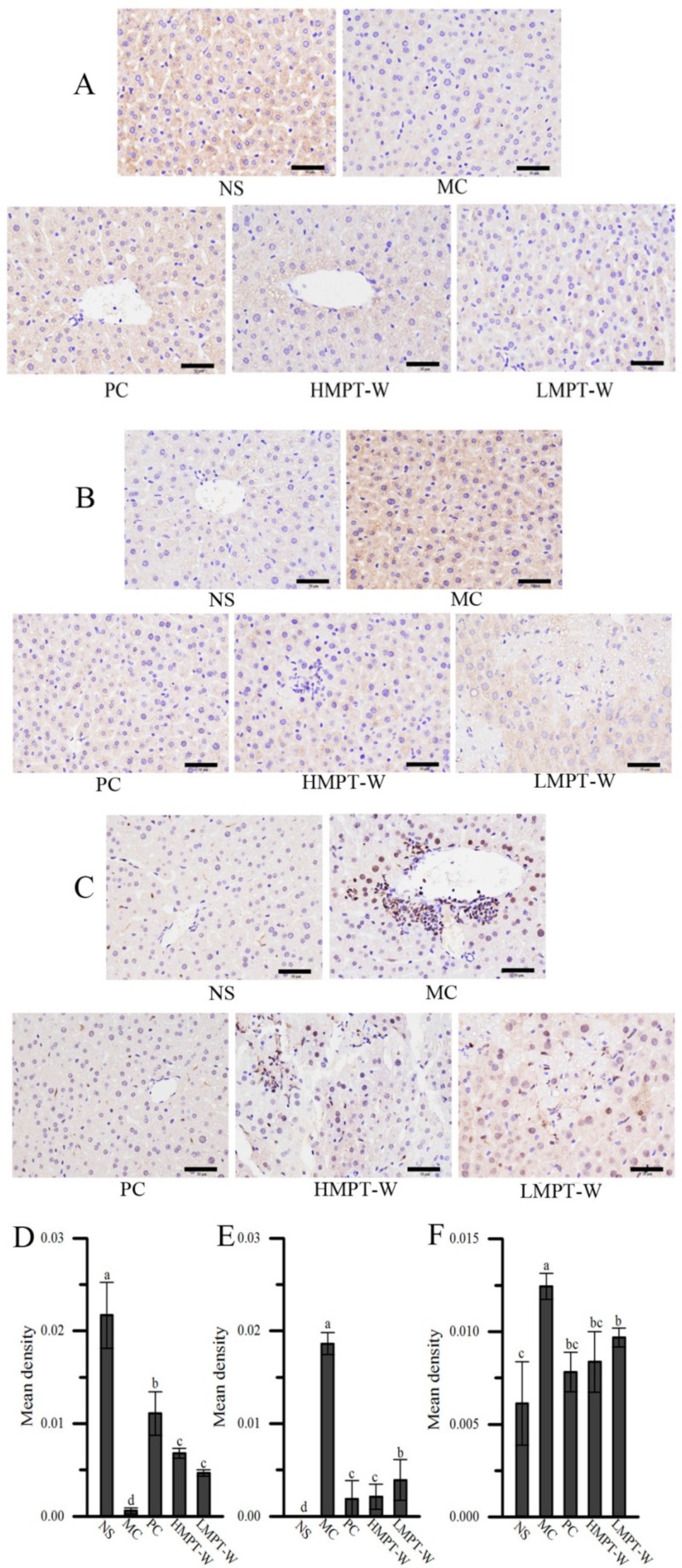
Immunohistochemistry analysis. Immunohistochemistry images of (**A**) IκBα, (**B**) IKK and (**C**) p-NF-κB p65; immunohistochemistry quantification of (**D**) IκBα, (**E**) p-IKK and (**F**) p-NF-κB p65. The values are reported as means ± SD. Bars with different letters (a, b, c, d, e) are significantly different (*p* < 0.05). p-NF-κB: phosphor-nuclear factor-kappa B, IκBα: NF-kappa-B inhibitor alpha, p-IKK: phosphor-inhibitor of nuclear factor kappa-B kinase alpha/beta.

**Table 1 ijms-20-04872-t001:** Mw, Mn and Mz of MPT-W.

Sample	Mn (Da)	Mw (Da)	Mz (Da)	Mw/Mn
MPT-W	1.13 × 10^5^	1.30 × 10^5^	1.49 × 10^5^	1.15

Mn: number-average molecular weight, Mw: weight-average molecular weight, Mz: Z-average molecular weight.

**Table 2 ijms-20-04872-t002:** Antioxidant activities in vitro.

Indexes	Sample	Concentrations (mg/L)						
		0	200	400	600	800	1000	1200
DPPH radicals	MPT-W	0	11.91 ± 0.84	24.87 ± 2.17	42.10 ± 1.70	63.84 ± 2.80	72.01 ± 2.05	76.81 ± 1.31
	Vc	0	61.07 ± 3.47	75.45 ± 2.07	87.24 ± 2.01	89.58 ± 1.07	91.04 ± 0.69	92.65 ± 1.13
Hydroxyl radicals	MPT-W	0	19.58 ± 1.98	38.10 ± 1.83	44.19 ± 1.20	54.84 ± 1.91	67.17 ± 1.97	74.11 ± 1.52
	Vc	0	41.26 ± 2.68	59.63 ± 0.89	69.45 ± 3.67	85.89 ± 2.17	92.75 ± 1.56	95.96 ± 0.84
Superoxide anion radicals	MPT-W	0	18.88 ± 1.29	36.12 ± 0.33	52.54 ± 1.169	57.49 ± 1.84	66.16 ± 1.71	72.59 ± 0.58
	Vc	0	32.69 ± 1.46	48.35 ± 0.79	65.50 ± 2.65	77.88 ± 1.30	82.84 ± 1.07	88.09 ± 1.49

Vc: ascorbic acid, DPPH: 1,1-diphenyl-2-picrylhydrazyl.

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
