# Peer review of "Mycelium Polysaccharides from Termitomyces albuminosus Attenuate CCl4-Induced Chronic Liver Injury Via Inhibiting TGFβ1/Smad3 and NF-κB Signal Pathways"

_ijms, 2019, doi:10.3390/ijms20194872_

Round 1
Reviewer 1 Report
The author has reported the potential antioxidant, anti-fibrosis and anti-inflammatory activities to prevent liver injury in CCl4-induced mice of the MPT-W from T. albuminosus.
The article is well written and the authors describe the various phases of the experimentation in detail.
There are a few things that I think need addressing in the manuscript that are detailed below:
The importance of MPT-W activities and highlights of the future implications or recommend direction for further research should be added. Please change in table the figure 2E To better understand and facilitate the reading I suggest to include the percentages of the scavenging activity, reported in 2.3 section, line 105-113, in a table. No reference to figure 9 (a-E) is given in the text. Please add. Line 306-307: please add the amount (mL) of water used. Line 308: what do the authors mean by “…by mixing with three volumes of 95% v/v ethanol..”? Check that the figures all have the same size and the same fontsAuthor Response
Reviewer 1 comments:
The author has reported the potential antioxidant, anti-fibrosis and anti-inflammatory activities to prevent liver injury in CCl4-induced mice of the MPT-W from T. albuminosus. The article is well written and the authors describe the various phases of the experimentation in detail. There are a few things that I think need addressing in the manuscript that are detailed below:
Thanks for your appreciation and affirmation of our manuscript.
The importance of MPT-W activities and highlights of the future implications or recommend direction for further research should be added.
Thanks for your kind comments. “The importance of MPT-W activities and highlights of the future implications or recommend direction for further research” had been added to the revised manuscript and marked in red.
Please change in table the figure 2E.
Thanks for your useful remind. The “figure 2E” had been removed from figure 2 as table 1.
To better understand and facilitate the reading I suggest to include the percentages of the scavenging activity, reported in 2.3 section, line 105-113, in a table
Thanks for your kind comments. The “figure 3” had been replaced by Table 2.
No reference to figure 9 (a-E) is given in the text. Please add.
Thanks for your useful remind. The figure 9(A-E) had been added to the revised manuscript, and marked in red.
Line 306-307: Please add the amount (mL) of water used
Thanks for your kind comments. The amount (mL) of water had been added to the revised manuscript, and marked in red.
Line 308: what do the authors mean by “…by mixing with three volumes of 95% v/v ethanol..”?
Thanks for your positive comments. To make it easier to understand this sentence, “the collected and concentrated supernatant was precipitated by mixing with three volumes of 95% v/v ethanol at 4 °C overnight” had been changed to “the collected and concentrated supernatant was mixed with three volumes of 95% v/v ethanol at 4 °C overnight to offer precipitate”, and marked in red in the revised manuscript.
Check that the figures all have the same size and the same fonts
Thanks for your kind comments. We had checked the size of figures and fonts.
The revised manuscript has been submitted to your journal. We look forward to receive your positive response.
Reviewer 2 Report
General comments: The manuscript contains too many grammatical and syntax errors, as well as poorly constructed sentences and poor choice of words. These editorial flaws render the manuscript poorly readable. It is recommended that authors seek for editing assistance from a professional English editor to look into the issues mentioned above.
Main comments:
1.) Inhibition of NF-kB activation is one of the main mechanisms being reported in this manuscript as a result of MPT-W administration. However, I see serious flaws in the experimental approaches for this part of the study, as listed below. These experimental flaws make the manuscript unacceptable for publication, mainly for reasons of accuracy and scientific soundness.
1.1. NF-kB activation is typically indicated by translocation of NF-KB p65 or p50 into the nucleus. Hence, the approach should have been that nuclear and cytosolic extracts were obtained and western blot conducted to monitor translocation of NF-kB p65. But the authors instead did immunoblotting using whole cell lysates. Increase in p65 in whole cell lysate doesnt show their translocation into the nucleus, and therefore does'nt mean activation of NF-kB. Moreover, increase in p65 in figure 9B, was not interpreted correctly. For MPT-W to inhibit NF-KB activation, authors were expected to show that NF-kB p65 nuclear translocation was inhibited or attenuated by MPT-W.
1.2. NF-kB activation is mainly indicated by phosporylation of IKKa/b. Authors should have measured phosphorylated form of IKKa/b (p-IKKa/b) and not IKKa/b (as shown in Fig. 9).
1.3. When conducting immunoblotting of phosphorylated proteins, phosphatase inhibitors should be added in addition to protease inhibitors. This was not indicated in section 4.10. Missing this important step in during western blotting, compromises the accuracy of results. Therefore, results on phosphorylated proteins as presented are highly questionable. Moreover, non-fat powdered milk is not recommended when blotting phosphorylated proteins (this is common knowledge in immunoblotting). Authors used NFDM as indicated in section 4.10.
Author Response
Reviewer 2 comments:
General comments: The manuscript contains too many grammatical and syntax errors, as well as poorly constructed sentences and poor choice of words. These editorial flaws render the manuscript poorly readable. It is recommended that authors seek for editing assistance from a professional English editor to look into the issues mentioned above.
Thanks for your kind comments. The manuscript had been edited MDPI English Editing Service, and the certification had been uploaded as Supplementary file.
Main comments:
1.) Inhibition of NF-kB activation is one of the main mechanisms being reported in this manuscript as a result of MPT-W administration. However, I see serious flaws in the experimental approaches for this part of the study, as listed below. These experimental flaws make the manuscript unacceptable for publication, mainly for reasons of accuracy and scientific soundness.
1.1. NF-kB activation is typically indicated by translocation of NF-KB p65 or p50 into the nucleus. Hence, the approach should have been that nuclear and cytosolic extracts were obtained and western blot conducted to monitor translocation of NF-kB p65. But the authors instead did immunoblotting using whole cell lysates. Increase in p65 in whole cell lysate doesnt show their translocation into the nucleus, and therefore does'nt mean activation of NF-kB. Moreover, increase in p65 in figure 9B, was not interpreted correctly. For MPT-W to inhibit NF-KB activation, authors were expected to show that NF-kB p65 nuclear translocation was inhibited or attenuated by MPT-W.
Thanks for your positive comments. NF-kB is usually activated after phosphorylation and the subsequent degradation of its inhibitor IkB. In our work, the aim of evaluating p-NF-kB p65 in total protein is to analyze the change of phosphorylation level of NF-kB p65. Meantime, some literatures have reported that p-NF-kB p65 is tested in total protein.
The related literatures were referenced as follows.
Khan, S.; Shehzad, O.; Jin, H. G.; Woo, E. R.; Kang, S. S.; Baek, S. W.; Kim, J.; Kim, Y. S. Anti-inflammatory mechanism of 15,16-epoxy-3α-hydroxylabda- 8,13(16),14-trien-7-one via inhibition of LPS-induced multicellular signaling pathways. J. Nat. Prod. 2012, 75, 67-71.
Valaskova, P.; Dvorak, A.; Lenicek, M.; Zizalova, K.; Kutinova-Canova, N.; Zelenka, J., Cahova, M.; Vitek, L.; Muchova, L. Hyperbilirubinemia in gunn rats is associated with decreased inflammatory response in LPS-mediated systemic inflammation. Int. J. Mol. Sci. 2019, 20, E2306.
Haghi Aminjan, H.’ Abtahi, SR.; Hazrati, E.; Chamanara. M; Jalili, M.; Paknejad, B. Targeting of oxidative stress and inflammation through ROS/NF-kappaB pathway in phosphine-induced hepatotoxicity mitigation. Life Sci. 2019, 232, 116607.
Macías-Pérez, J. R.; Aldaba-Muruato, L. R.; Martínez-Hernández, S. L.; Muñoz-Ortega, M. H.; Pulido-Ortega, J.; Ventura-Juárez, J. Curcumin provides hepatoprotection against amoebic liver abscess induced by entamoeba histolytica in hamster: involvement of Nrf2/HO-1 and NF-κB/IL-1β signaling pathways. J. Immunol. Res. 2019, 2019, 7431652.
Xiao, Q.; Zhang, S.; Yang, C.; Du, R.; Zhao, J.; Li, J.; Xu, Y.; Qin, Y.; Gao, Y.; Huang, W. Ginsenoside Rg1 ameliorates palmitic acid-induced hepatic steatosis and inflammation in hepG2 cells via the AMPK/NF-κB pathway. Int. J. Endocrinol. 2019, 2019, 7514802.
Ji, Y.; Ge, Y.; Xu, X.; Ye, S.; Fan, Y.; Zhang, J. Vildagliptin reduces stenosis of injured carotid artery in diabetic mouse through inhibiting vascular smooth muscle cell proliferation via ER stress/NF-κB pathway. Front. Pharmacol. 2019, 10, 142.
Meng, B.; Zhang, Y.; Wang, Z.; Ding, Q.; Song, J.; Wang, D. Hepatoprotective effects of Morchella esculenta against alcohol-induced acute liver injury in the C57BL/6 mouse related to Nrf-2 and NF-κB signaling. Oxid. Med. Cell. Longev. 2019, 2019, 6029876.
1.2. NF-kB activation is mainly indicated by phosporylation of IKKa/b. Authors should have measured phosphorylated form of IKKa/b (p-IKKa/b) and not IKKa/b (as shown in Fig. 9).
Thanks for your positive comments. IKKα/β had been removed from this manuscript, and p-IKKα/β had been evaluated and added to the revised manuscript, marked in red.
1.3. When conducting immunoblotting of phosphorylated proteins, phosphatase inhibitors should be added in addition to protease inhibitors. This was not indicated in section 4.10. Missing this important step in during western blotting, compromises the accuracy of results. Therefore, results on phosphorylated proteins as presented are highly questionable. Moreover, non-fat powdered milk is not recommended when blotting phosphorylated proteins (this is common knowledge in immunoblotting). Authors used NFDM as indicated in section 4.10.
Thanks for your positive comments. The phosphatase inhibitor had been added to the revised manuscript, marked in red.
Some literatures have reported that non-fat powdered milk is used when blotting phosphorylated proteins.
Khan, S.; Shehzad, O.; Jin, H. G.; Woo, E. R.; Kang, S. S.; Baek, S. W.; Kim, J.; Kim, Y. S. Anti-inflammatory mechanism of 15,16-epoxy-3α-hydroxylabda- 8,13(16),14-trien-7-one via inhibition of LPS-induced multicellular signaling pathways. J. Nat. Prod. 2012, 75, 67-71.
Hsia, C. H.; Velusamy, M.; Jayakumar, T.; Chen, Y. J.; Hsia, C. W.; Tsai, J. H.; Mechanisms of TQ-6, a novel ruthenium-derivative compound, against lipopolysaccharide-induced in vitro macrophage activation and liver injury in experimental mice: The crucial role of p38 MAPK and NF-κB signaling. Cells, 2018, 7, 217.
Al Zoubi, S.; Chen, J.; Murphy, C.; Martin, L.; Chiazza, F.; Collotta, D.; Linagliptin attenuates the cardiac dysfunction associated with experimental sepsis in mice with pre-existing type 2 diabetes by inhibiting NF-κB. Front. Immunol. 2018, 9, 2996.
Ji, Y.; Ge, Y.; Xu, X.; Ye, S.; Fan, Y.; Zhang, J. Vildagliptin reduces stenosis of injured carotid artery in diabetic mouse through inhibiting vascular smooth muscle cell proliferation via ER stress/NF-κB pathway. Front. Pharmacol. 2019, 10, 142.
Xiao, Q.; Zhang, S.; Yang, C.; Du, R.; Zhao, J.; Li, J.; Xu, Y.; Qin, Y.; Gao, Y.; Huang, W. Ginsenoside Rg1 ameliorates palmitic acid-induced hepatic steatosis and inflammation in hepG2 cells via the AMPK/NF-κB pathway. Int. J. Endocrinol. 2019, 2019, 7514802.
Zheng, H.; Wang, X.; Zhang, Y.; Chen, L.; Hua, L.; Xu, W. Pien-Tze-Huang ameliorates hepatic fibrosis via suppressing NF-κB pathway and promoting HSC apoptosis. J. Ethnopharmacol. 2019, 244, 111856.
The revised manuscript has been submitted to your journal. We look forward to receive your positive response.
Reviewer 3 Report
There is an interesting paper, however, a significant improvement of data presentation and interpretation is needed.
The manuscript should be corrected by a native speaker familiar with medical/biological sciences.
Introduction should be improved. There is not well documented that T. albuminosus can be used to treat listed diseases.
The figure legends must be improved. A figure legend must contain all information necessary for readers to understand presented data: e.g. the legend of Fig.4 should contain explanation what kind of samples were used to assess AST activity: it is not clear what “NC, MC, PC, HMPT-W, LMPT-W” mean. A reader to understand this figure must go to other parts of the manuscript to find out what kind of samples were analyzed. The meaning of letters “a, b, c, d, e” should also be explained under figure legends.
It is not clear why CYP2E1 was chosen. Some explanation under Results section is needed.
P values should be also presented, it is not enough it say that p is lower than 0.05. Values lower than 0.05 indicate statistically significant differences but do not indicate a strength of a specific difference (p<0.05, p<0.01, p<,0.001 indicate the strength of specific differences).
It is hard to find references in the Methods section and not all methods are precisely described, e.g. ROS assay should be described.
Author Response
Reviewer 3 comments:
There is an interesting paper, however, a significant improvement of data presentation and interpretation is needed.
Thanks for your appreciation and affirmation of our manuscript.
The manuscript should be corrected by a native speaker familiar with medical/biological sciences.
Thanks for your kind comments. The manuscript had been edited MDPI English Editing Service, and the certification had been uploaded as Supplementary file.
Introduction should be improved. There is not well documented that T. albuminosus can be used to treat listed diseases.
Thanks for your positive comments. The references had been added to the revised manuscript, marked in red.
The figure legends must be improved. A figure legend must contain all information necessary for readers to understand presented data: e.g. the legend of Fig.4 should contain explanation what kind of samples were used to assess AST activity: it is not clear what “NC, MC, PC, HMPT-W, LMPT-W” mean. A reader to understand this figure must go to other parts of the manuscript to find out what kind of samples were analyzed. The meaning of letters “a, b, c, d, e” should also be explained under figure legends.
Thanks for your kind comments.
The sample (serum/liver), used to assess indexes, had been added to all figure legends, and marked in red in the revised manuscript.
To understand well figures, the complete spellings of the important abbreviations had been added to all figure legends, and marked in red in the revised manuscript.
The letters “a, b, c, d, e” had been explained in figure legends.
It is not clear why CYP2E1 was chosen. Some explanation under Results section is needed.
Thanks for your kind comments. CYP2E1 had been explained, and marked in red in the revised manuscript.
P values should be also presented, it is not enough it say that p is lower than 0.05. Values lower than 0.05 indicate statistically significant differences but do not indicate a strength of a specific difference (p<0.05, p<0.01, p<0.001 indicate the strength of specific differences).
Thanks for your positive comments. To well show a specific difference, all P values had been added to the revised manuscript, and marked in red.
It is hard to find references in the Methods section Sand not all methods are precisely described, e.g. ROS assay should be described.
Thanks for your kind comments. The reference about ROS assay had been added to the revised manuscript, and marked in red.
The revised manuscript has been submitted to your journal. We look forward to receive your positive response.
Reviewer 4 Report
Zhao et al report polysaccharides from Mycelium of Termitomyces albuminosus have antioxidant activities in vitro and protect against CCl4-induced liver injury in vivo. They used various molecular techniques to show the antioxidant and liver protective effects of MPT. The concluded that liver protective activity of MPT was from antioxidant and anti-inflammatory effects. Overall, data is convincing and supports the conclusion. However, English writing needs extensive editing. Following is suggestion for authors.
Number of replications in all the data shall be given. Especially, when SD is surprisingly small. What is the "PC"? The identity of PC shall be given. Rationale for CYP2E1 measurement for evaluating CCl4 liver toxicity shall be given first. 2-desoxyribose -> 2-deoxyribose Figure 6 is too unclear to support the conclusion of authors. Dihydroethidium method shall be supported by a reference. Figure 6 y-axis needs unit. How was CYP2E1 measured in pmol/mL. There is no method describing it. Statistics, Duncan test is usually used as a post hoc test after ANOVA. ANOVA is not mentioned. Methods need more details.
Author Response
Reviewer 4 comments:
Zhao et al report polysaccharides from Mycelium of Termitomycesalbuminosus have antioxidant activities in vitro and protect against CCl4-induced liver injury in vivo. They used various molecular techniques to show the antioxidant and liver protective effects of MPT. The concluded that liver protective activity of MPT was from antioxidant and anti-inflammatory effects. Overall, data is convincing and supports the conclusion. However, English writing needs extensive editing. Following is suggestion for authors.
Thanks for your kind comments. The manuscript had been edited MDPI English Editing Service, and the certification had been uploaded as Supplementary file.
Number of replications in all the data shall be given. Especially, when SD is surprisingly small.
Thanks for your kind comments. Number of replications in all the data had been added to statistical analysis section.
What is the "PC"? The identity of PC shall be given.
Thanks for your useful remind. “PC” was abbreviation of positive control, and had been explained in method section and all figure legends.
Rationale for CYP2E1 measurement for evaluating CCl4 liver toxicity shall be given first.
Thanks for your positive comments. CYP2E1 was analyzed using commercial kits according to the instructions. The related information had been added to materials and methods section
Figure 6 is too unclear to support the conclusion of authors. Dihydroethidium method shall be supported by a reference. Figure 6 y-axis needs unit.
Thanks for your useful remind.
The resolution of figure 6 had been improved.
The reference about ROS assay had been added to the revised manuscript, and marked in red.
Figure 6 y-axis needs unit had been added.
How was CYP2E1 measured in pmol/mL. There is no method describing it.
Thanks for your kind comments. CYP2E1 was analyzed using commercial kits according to the instructions. The related information had been added to materials and methods section
Statistics, Duncan test is usually used as a post hoc test after ANOVA. ANOVA is not mentioned.
Thanks for your useful remind. Statistical analysis section had been rewritten, and marked in red in the revised manuscript.
Methods need more details.
Thanks for your kind comments. Method had been checked, and the related reference and explanation had been added to the revised manuscript.
The revised manuscript has been submitted to your journal. We look forward to receive your positive response.
Round 2
Reviewer 3 Report
The manuscript was improved according to previous comments. No additional comments